# Review of Under-Recognized Adjunctive Therapies for Cancer

**DOI:** 10.3390/cancers14194780

**Published:** 2022-09-29

**Authors:** Mary E. Money, Carolyn M. Matthews, Jocelyn Tan-Shalaby

**Affiliations:** 1Department of Medicine, University of Maryland School of Medicine, 665 W Baltimore Street S, Baltimore, MD 21201, USA; 2Meritus Medical Center, 11116 Medical Campus Rd., Hagerstown, MD 21742, USA; 3Texas Oncology, PA and Charles A. Sammons Cancer Center, 3410 Worth St., Suite 400, Dallas, TX 75246, USA; 4Gynecologic Oncology, Baylor Sammons Cancer Center, 3410 Worth St., Suite 400, Dallas, TX 75246, USA; 5Department of Medicine, University of Pittsburgh School of Medicine, 3550 Terrace St., Pittsburgh, PA 15213, USA; 6Department of Medicine, Veteran Affairs Pittsburgh Healthcare System, 4100 Allequippa St., Pittsburgh, PA 15240, USA

**Keywords:** cancer, green tea, adjunctive cancer therapy, melatonin, stress reduction techniques, mindfulness, yoga, Tai Chi, turmeric, statin therapy, metformin, exercise effect on cancer, repurposed drugs for cancer treatment, diet influence on cancer, fasting prior to chemotherapy, ketogenic diet, circadian rhythm effect on immunity, improved sleep effect on cancer, integrative oncology

## Abstract

**Simple Summary:**

This review presents cancer and primary care providers with an overview of underappreciated adjunctive measures that may improve their patients’ quality of life and survival. This is not a comprehensive review but adjunctive recommendations which may be easily addressed by the provider and acceptable to the cancer patient. These include exercise, diet, stress-reduction techniques, recognition and management of sleep problems, advice on smoking cessation, and use of selective nutraceuticals and pharmaceuticals as adjuvants. In addition, patients may be more compliant if suggestions and referrals are made by their trusted providers.

**Abstract:**

Patients and providers may not be aware that several adjunctive measures can significantly improve the quality of life, response to treatment, and possibly outcomes for cancer patients. This manuscript presents a review of practical under-recognized adjunctive therapies that are effective including exercise; stress-reduction techniques such as mindfulness, massage, yoga, Tai Chi, breathing exercises; importance of sleep quality; diet modifications such as calorie restriction at the time of chemotherapy and avoidance of high carbohydrate foods; supplements such as aspirin, green tea, turmeric, and melatonin; and repurposed prescription medications such as metformin and statins. Each recommendation should be tailored to the individual patient to assure no contraindications.

## 1. Introduction

After a patient has been diagnosed with cancer, treatment is usually initiated with chemotherapy or radiation therapy, and adjunctive therapy may not be considered. An adjunctive therapy is one that may improve quality of life and/or survival and should be unlikely to cause harm. A recent survey of members of the American Society of Clinical Oncologist identified that most respondents agreed with the importance of exercise, weight management and diet recommendations for cancer patients. However, only 19–23% of oncologists refer patients to an exercise program [1]. A 2017 systematic review concluded that “exercise is beneficial before, during, and after cancer treatment, across all cancer types, and for a variety of cancer-related impairments”. Moderate-to-vigorous exercise is the best level of exercise intensity to improve physical function and mitigate cancer-related impairments [2]. Now, multiple medical societies consider exercise as “medicine”. Unfortunately, unless the individual had previously established an exercise routine, just advising them may not be sufficient and a referral to an exercise program is necessary.

This review highlights the importance of all types of exercise and other unappreciated adjunctive therapies such as stress-reduction techniques including meditation or mindfulness, breathing exercises, massage, good sleep hygiene, melatonin, turmeric, green tea, diet modifications, and repurposed medications for cancer patients. Some cancers are known to have a high recurrence rate, typically related to the presence of residual cancer stem cells [3], which can be modulated by some of the therapies discussed here. It is important for the oncologist as well as the primary care provider to be knowledgeable regarding these adjunctive therapies be willing to recommend them as well as using targeted chemotherapy agents. 

## 2. Exercise as Cancer Therapy

### 2.1. Exercise Recommendations

The recently published American Society of Clinical Oncology guidelines now clearly recommends that oncologists should encourage patients undergoing active treatment with curative goal to conduct regular aerobic and resistance exercise because of the beneficial evidence [4]. Specific guidance for patients with advanced cancers necessitates further research [4]. Another systematic review and meta-analysis demonstrated that exercise could significantly reduce mortality in patients with cancer [5]. The intensity and quantity of exercise were significant factors that affected the immune system, cancer cell survival, and angiogenesis [6]. Immune lymphocytic natural killer cells numbers are rapidly increased with 30 min of moderate exercise [7]. Tumor cells survive in poorly oxygenated tissue and exercise can improve oxygenation through increased cardiac output, resulting in better efficacy of therapy. Prior studies have indicated that vigorous exercise of six or more metabolic equivalents per hour was more effective than a less intense exercise of longer duration. Encouraging patients to achieve ≥150 min weekly of moderate or 75 min of vigorous exercise along with resistance exercise for all major muscle groups is recommended [7].

Unfortunately, surveys have identified that barriers exist for patients to accomplish these recommendations and providing an exercise DVD simultaneously improved compliance among breast cancer patients [8]. An 11-week randomized trial involving head and neck cancer (HNC) patients receiving chemo-radiotherapy comparing structured to conventional exercise recommendations showed a statistical improvement in quality of life, less fatigue and maintenance of functional capacity among the structured group [9].

### 2.2. Tai Chi, Yoga, and Baduanjin as a Form of Exercise

Since some patients may not be physically able to perform aerobic exercise or resistance training, Tai Chi, yoga, or Baduanjin exercises may be more feasible alternatives to suggest. A systematic review evaluating Tai Chi identified benefits for fatigue and sleep quality among cancer patients, but there was insufficient evidence on long term outcomes [10]. Another exercise option is also Baduanjin, a tradition Chinese Qigong exercise that consists of eight easy movements of the body, which in a systematic review and me-ta-analysis of randomized control trials, has been found to improve cancer fatigue, sleep, and quality of life among cancer patients [11].

Yoga and Tai Chi have numerous benefits including stress reduction and physical conditioning for cancer patients. Yoga therapy has been demonstrated to reduce fatigue [12], improve quality of life [13], and reduce daily pain among patients with metastatic breast cancer [14]. These findings were also identified in a recent meta-analysis among breast cancer patients taking yoga to controls who were not taking yoga, showing yoga improved QoL, reduced stress, depression, fatigue, anxiety, and pain severity [15]. A similar meta-analysis focusing on Tai Chi identified a positive short-term effect on cancer related fatigue in some cancer patients the longer it was performed [16].

## 3. Stress-Reduction Practices

Receiving a cancer diagnosis is a major emotional shock for a patient and providers should acknowledge the stress reaction that occurs among patients. To help patients manage this stress, besides incorporating some form of regular exercise as discussed above, providers should encourage patients to explore non-physical options such as meditation or mindfulness practices, massage, art or music therapy, acupuncture, or mindful breathing practices. Each of these has beneficial effects and one or more may appeal to the patient. Providers should be willing to refer the patient for counseling or therapy as requested by the patient’s preference. Stress associated with a cancer diagnosis may activate both the sympathetic nervous system and the hypothalamic-pituitary-adrenal axis. These activations may result in the release of glucocorticoid hormones that contribute to the progression of cancer [17].

In the 1970s, Kabat-Zinn [18] established the first standardized mindfulness-based intervention, called mindfulness-based stress reduction (MBSR). Currently, many similar programs are available on the internet. This training encourages individuals to live in the present, purposely pay attention, and be nonjudgmental to the experience. A recent systematic review and meta-analysis of 29 randomized controlled trials involving 3476 patients found that patients with mindfulness training experienced significant improvements in quality of life and reductions in stress, anxiety, fatigue, and depression, compared to controls [19]. Additional studies are currently underway to assess the benefits of mindfulness and its effect on cancer immunity [20]. However, there is an abundance of evidence that supports the initial benefits of mindfulness on quality-of-life improvements for patients faced with cancer [21,22]. Some cancer centers have begun incorporating integrative practices to respond to the unmet emotional needs of cancer patients [22,23]. Examples of practices that are incorporated after a consultation with patients may include massage, acupuncture, MBSR counseling, art and music therapy. Clinical practice guidelines on the evidence-based use of such practices during and after breast cancer treatment is now available for reference [24]. Both acupuncture treatment and massage therapy were each found to significantly reduce stress and pain among cancer patients [23]. Although there are limited references regarding the effectiveness of art or music therapy, a recent systematic review concluded benefits of both therapies for breast cancer patients for improving quality of life and emotional well-being [24,25]. There are several different techniques on mindful breathing practices that have been shown to help reduce stress or anxiety and depression. Examples include diaphragmatic breathing [26], Sudarshan Kriya Yoga (SKY) breathing [27], and “mindful breathing” a technique that requires a patient to sit upright, and for five minutes, three times daily focus only on their breathing [28]. This simple practice of recentering their mind if it wanders has been shown to significantly reduce stress.

## 4. Management of Sleep Disturbances as an Adjunctive Cancer Therapy

Although management of sleep disorders may not be considered an adjunctive therapy, any therapy that aids the primary goal of helping to treat the cancer patient is an adjunctive therapy. Unfortunately, many providers may not recognize the importance of sleep regarding its effect on the immune system and therefore its effect on cancer survival.

A systematic review by Santoso identified insomnia, hypersomnolence, and sleep-related breathing disturbances among HNC patients were common before, during and after treatment [29]. Reviews using the Pittsburgh Sleep Quality Index (PSQI) sleep study survey, identified that 69% of patients with cancer experienced poor sleep [30], and close to a third of HNC patients have either persistent or worsening sleep problems [31]. Obstructive sleep apnea occurs more often in HNC than other cancers both before and after treatment [32,33,34]. 

The importance of sleep in the survival of patients with cancer has not been adequately studied, but a recent review of how sleep affects the immune system supported the hypothesis that poor sleep could lead to unfavorable outcomes [35].

The 2017 Nobel Prize in Physiology and Medicine was awarded for the discovery of the molecular mechanisms that control the circadian rhythm [36], which is critical for normal cellular function. Patel and Kondratov [37] recently reviewed the importance of circadian clock genes and their relationship to the metabolic pathways that control cancer development. They showed that a high incidence of cancer was found in night-shift workers. That finding resulted in the 2019 designation of night-shift work as a Group 2A carcinogen by the International Agency for Research on Cancer [38]. Examples of night-shift work-related risks included increases in breast cancer among women [39] and prostate cancer among men [40]. The disproportionate increase in cancer among night-shift workers is potentially related to a disruption of melatonin activities [41].

Analyses of the Circadian Gene Database have estimated that approximately 2000 genes that may be influenced by the circadian rhythm could affect cancer [42]. The EPIdemiological study of Prostate CAncer (EPICAP) was a population-based, case–control study specifically designed to investigate the role of environmental and genetic factors in prostate cancer. A recent analysis of circadian gene variants in the men enrolled in EPICAP identified specific circadian genes associated with aggressive prostate cancer among night workers [43]. Liu et al. [44] presented a comprehensive analysis of how some cancers affected the circadian clock.

Patients often complain of disordered sleep after initiating chemotherapy. Savard et al. confirmed that progressive sleep impairments occurred with repeated chemotherapy, based on wrist actigraphy recordings from patients with breast cancer [45]. In a prospective cohort study to determine whether chemotherapy could impair melatonin production, Li et al. [46] examined 180 women who underwent adjuvant chemotherapy for breast cancer. They examined first morning urinary melatonin levels and sleep–wake activity, rhythms, and patterns. They found disruptions in the sleep–wake rhythm and a reduction in the urinary melatonin level.

After sleep apnea is excluded as a cause of poor sleep, and is corrected with appropriate measures, the first-line treatment is typically cognitive behavior therapy [47], but this approach may be time-consuming, and patients may not be receptive. Some studies have shown that administering melatonin could provide benefit by helping reset the circadian rhythm and improve sleep [48]. A randomized, placebo-controlled trial for testing melatonin treatment demonstrated that melatonin improved sleeping behavior among breast-cancer survivors [49]. The dose of melatonin that a patient needs to have restorative sleep may vary from 0.5 mg to 20 mg depending on how melatonin is metabolized by that individual. A recent study found that the timing of administration of melatonin was important. The survival benefit among patients with advanced non-small-cell lung cancer was only seen with melatonin administered in the evening and among those whose sleep had normalized [50]. Since the secretion of melatonin decreases as a person grows older, a recent review concluded that patients should begin with an initial melatonin trial to correct impaired sleep to improve quality of life. If ineffective, then it is important to prescribe other medications to optimize the sleep quality in the cancer patient.

## 5. Melatonin, a Potential Chemotherapy Agent

Melatonin is mainly produced by the pineal gland, from tryptophan, in response to darkness. Melatonin is released as it is synthesized, and it contributes to regulating homeostatic metabolic rhythms to protect the body from disease development. It has been shown that melatonin regulates sleep–wake cycles, the immune system, bone homeostasis, and arterial blood pressure, provides potential neuroprotection, and acts as an antioxidant [48].

As an antioxidant, melatonin has been shown to (1) stimulate antioxidant synthesis, (2) scavenge reactive oxygen species and reactive nitrogen species, (3) reduce free-radical generation, (4) generate a radical scavenger cascade, (5) enhance other antioxidant activities, and (6) regulate antioxidant enzymes [48]. In addition, melatonin is an anti-inflammatory agent, has anti-angiogenetic effects, and potentiates cancer cell apoptosis [51]. It has also been shown to potentiate the actions of other antioxidants and chemotherapy agents [41,48].

Extensive studies have focused on the potential beneficial effect of melatonin in HNC patients [53,54,55,56]. González et al. [57] reviewed 17 clinical trials and analyzed the antiangiogenic effects of melatonin. Positive effects included longer survival times, less severe side effects, and better quality of life. Melatonin dosages were 10–20 mg, given 1 h before bedtime.

A recent study using 20 mg of melatonin after neoadjuvant chemotherapy for locally advanced oral squamous cell carcinoma found a “decrease in the expression of miR-210 and CD44 followed by a decrease in the percentage of the residual tumor but not significant (*p* = 0.114) [58].

A comprehensive review of the clinical trials using melatonin as adjunctive therapy and the biological effects of melatonin has recently been published and discusses the limitations that need to be addressed going forward with clinical trials utilizing this medication for cancer therapy [45,46,49,59].

The safety profile of high doses of melatonin has recently been reviewed. The most common adverse events were tiredness, dizziness, drowsiness, fever, headache, and diarrhea which were also present in study control groups [60].

## 6. Diet 

### 6.1. Overview of Recommendations for Diet after Cancer Diagnosis

While randomized prospective dietary trials are limited, there is some evidence that suggests patients diagnosed with cancer can improve their future health by following a careful well-balanced diet. A recent review demonstrated this effect by evaluating food frequency questionnaires. They identified significant reductions in the risks of 10-year cancer-specific and all-cause mortality among patients with cancer that followed dietary recommendations, compared to controls. Diets were evaluated with the Healthy Eating Index (HEI)-2015, the Alternative HEI-2010, the alternate Mediterranean Diet, or the Dietary Approaches to Stop Hypertension scores [61]. Table 1 outlines the recent basic dietary recommendations for reducing the risk of cancer recurrence [62].

### 6.2. Ketogenic Diet Therapy for Cancer

The benefit of a ketogenic diet may be due to the Warburg effect in cancer cells [63,64], which results in the predominant utilization of glycolysis and little or no ability to utilize ketones. It also has a physiological effect on the nervous system, circadian clock, metabolism, immune system and may increase the diversity of the microbiome [63,64]. HNC patients had difficulty complying with a ketogenic diet in a phase 1 trial [65], but an interim analysis of the KETOCOMP study (NCT 02516501) showed a ketogenic diet in HNC patients helped maintain body weight and skeletal mass [66]. A new ketogenic regimen trial for stage IV cancer patients, which varied the carbohydrate restriction from 10 g daily for the first week, 20 g daily for weeks 2–12 and subsequently to 30 g daily, did result in a modest improvement in survival among some patients [67].

Although Romer et al. concluded that evidence is lacking on the clinical efficacy of ketogenic diets for cancer due to the poor adherence, heterogeneous results, and methodological limitations of the trials reviewed [68], Jemal et al. recently reviewed the therapeutic potential of the ketogenic diet for patients with breast cancer [69].

Human studies confirming benefits are limited by small patient numbers, high patient drop-out rates, or retrospective case-study designs. It is reasonable, however, to review the benefits of this diet among cancer patients and provide a referral to a dietician if requested.

### 6.3. Short-Term Fasting Prior to Chemotherapy

Recent studies and reviews have demonstrated that short-term fasting, either immediately before each cycle of chemotherapy or intermittently, reduced chemotherapy toxicity [70,71,72,73,74]. Intermittent fasting during chemotherapy could potentially improve treatment-related side effects, insulin sensitivity, chemotherapy effectiveness, and quality of life [75]. Further research is ongoing, but patients should be at least advised about this option and encouraged to consider short-term fasting, when motivated. An example of fasting before a chemotherapy cycle would be a restriction of ≤200 calories for each 24 h cycle, during the period from 48 h before to 24 h after chemotherapy [73].

### 6.4. Duration of Fasting and Cancer Recurrence

Marinac et al. reviewed the effects of different durations of nightly fasting on breast cancer recurrence among 2413 women without diabetes. They found that fasting less than 13 h overnight was associated with a higher risk of recurrence [76].

## 7. Smoking

### Effect of Smoking and Response to Chemotherapy

Patients should be encouraged to stop smoking for multiple reasons. Numerous studies have shown that smoking has an adverse effect on the efficacy of chemotherapy [77]. Patients with lung cancer who continue to smoke have a reduced life expectancy compared to patients who stop smoking [78]. In addition, there is an increased risk of a second cancer among active smokers [79].

## 8. Green Tea

The consideration of green tea (GT) as an adjunctive measure is based upon its recognized health benefits since very few trials have been completed using GT because of the difficulty confirming the quantity of tea ingested or the lack of uniformity of green tea extracts (GTE). The information presented here is the current knowledge about how green tea affects cancer cells and has the potential to be effective for cancer treatment.

### 8.1. Background

Green tea (GT) contains elevated levels of catechins also known as soluble polyphenols, which represent between 30 and 42% of the dry weight [80]. For centuries, Eastern cultures have recognized the health benefits of GT, and recent studies have identified its ability to kill cancer cells and prevent cancer without harming normal cells [81]. However, Western physicians have not widely adopted GT for use in therapy regimens. Several epidemiology reviews have confirmed an inverse relationship between GT consumption and cancer incidence. A cup of GT contains 10x the amount of (-)-epigallocatechin gallate (EGCG, a bioactive catechin) found in a cup of coffee [82]. Research studies have demonstrated that tea catechins peak in the plasma within 1–5 h and have a half-life of 2–10 h [83]. Multiple servings of GT daily enhance tissue accumulation by 4–9x [84]. The importance of GT consumption is reflected in the ancient Chinese proverb “Better to be deprived of food for three days, than tea for one” [81]. Since GT side effects are minimal, this readily available drink could benefit anyone’s health, particularly patients with cancer.

### 8.2. Mechanism of GT Effect on Cancer Cells [85,86,87,88,89,90,91]

In 1987, it was first recognized that GT catechins inhibited the lung cancer tumor promoter in the *TNF-**α* gene [84]. The benefit of GT is thought to be derived from EGCG and a combination of other catechins. There are comprehensive reviews available on the potential molecular targets and signaling pathways involved in the GT effects on cancer cells [85,86,87,88,89,90]. Most of these effects are summarized in Table 2. 

Ruhul Amin et al. identified that the combination of EGCG and resveratrol synergistically increased apoptosis in xenografted head and neck tumors in nude mice by inhibiting AKT-mTOR signaling in vitro and in vivo [91].

### 8.3. Human Studies: Evidence of Green Tea Effects on Specific Cancers

Prospective and epidemiological studies have suggested that GT is effective in preventing or delaying recurrence of the certain cancers [92,93]. For example:

**Stages I and II breast cancer:** Compared to patients who consumed ≤4 cups of GT/day, those who consumed ≥5 cups/day had fewer recurrences (16.7% vs. 24.3%; *p* < 0.05) over seven years, and a longer average disease-free period (3.6 vs. 2.6 years) [92].

**High-grade prostate intra-epithelial neoplasia:** A “proof-of-principal” double blind, placebo-controlled, one year study (N = 60 men) showed that, among 30 men taking (3) 200 mg GT catechin capsules daily, only one prostate cancer was diagnosed, compared to nine invasive cancers diagnosed in the placebo arm [94]. 

**Advanced premalignant lesions of the oral cavity and larynx:** Treatment with green tea polyphenon E (200 mg 3x day) combined with escalating dose of EGFR-tyrosine kinase inhibitor (erlotinib) in a phase 1 B study found the combination was well tolerated and 17/21 patients showed pathological improvement and increased cancer free survival [95].

### 8.4. Safety Concerns

Although green tea extract (GTE) or a bolus dose of EGCG was causally associated with hepatic toxicity, no toxicity was observed with GT beverages across “a wide range of intakes and conditions” [83]. GTE are not all the same, and catechin profiles differ depending upon the manufacturing process [96]. A GTE limit of 300 mg/day is advised, due to hepatic side effects, though a recent study on lupus found no adverse reactions after administering 1000 mg/day EGCG [97]. Additionally, mild hypertension was associated with GT, due to the caffeine content (¼ the caffeine content of black tea). EGCG is also a potential P-glycoprotein substrate, and thus, it may influence the availability of drugs that require this transport enzyme, such as digoxin [98]. Therefore, a drug reaction evaluation is recommended. However, GT exhibited anticancer effects that were synergistic with numerous chemotherapy agents, in vitro [99].

Japanese loose-leaf GT has almost twice the EGCG content as Chinese tea [100]. The majority of EGCG is extracted in 160–180 °F water after 3–4 min. Longer brewing may cause catechin degradation [101]. It is universally accepted that brewing loose leaf tea provides more benefits than tea bags.

Evaluation of potential drug–drug reaction with GT should be performed prior to recommending green tea consumption to a patient. If there is no contraindication, then the interested patient should be encouraged to consume at least 1000 cc of green tea daily throughout the day for the optimal antineoplastic benefit.

## 9. Curcumin/Turmeric

For centuries, Asian cultures have recognized the medicinal properties of turmeric (*Curcuma longa)*, a cooking spice. In cancer, turmeric has antioxidant, anti-inflammatory, anti-angiogenic, chemo-sensitizing, and apoptotic properties [102,103]. Multiple studies have demonstrated the effects of turmeric against cancer cells in vitro [104,105,106], and a recent review by Kabir discusses the effectiveness of newer formulations of turmeric against multiple cancer types including HNC [107]. Because turmeric absorption is poor, recent research has focused on how to incorporate turmeric into chemotherapy [108,109,110]. Farghadani and Naidu reviewed recent studies and human trials on curcumin effects in breast cancer [111], and Chen reported that curcumin analog HO-3867 activates JNK1/2 signaling in human oral squamous cell cancer cells [112].

Several meta-analyses and controlled trials have identified the effectiveness of different formulations of turmeric for control of treatment induced oral mucositis among HNC patients [113,114,115]. 

There are no known significant side effects of curcumin, rarely patients may rarely report various complaints, such as flatulence, diarrhea, nausea, constipation, tongue redness, tachycardia, or yellow stools. However, curcumin may interact with cardiovascular agents, antibiotics, antidepressants, anticoagulants, chemotherapeutic drugs, and antihistamines [116]. It is usually recommended for better absorption for it to be taken with food containing fat. 

## 10. Aspirin

The anticancer effect of aspirin was first recognized by Gasic et al., in the 1970s [117]. Patrono and Rocca reviewed studies on aspirin for preventing and treating gastrointestinal cancer. They demonstrated that aspirin significantly reduced cancer incidence and mortality [118]. Multiple prospective studies are currently ongoing on the use of aspirin as adjunctive therapy for cancer, because its effects include blunting proliferative signaling, restoring growth suppressors, modulating immune cells, inhibiting telomerase (which reduces cellular immortality), reducing inflammation, impeding invasion and metastasis, restricting angiogenesis, inhibiting genome instability, and reversing energy metabolism reprogramming [119].

Elwood recently completed an extensive review of published observational studies where aspirin was administered to cancer patients and identified a potential 20% reduction in mortality among aspirin users [120]. However, a Danish national cohort study of patients with head and neck squamous cell cancer did not find an overall survival benefit from aspirin use [121]. As with any therapy, the potential benefit of aspirin regarding survival in patients with cancer versus its potential risk should be carefully assessed on an individual basis.

## 11. Metformin

Metformin is being extensively studied as adjunctive therapy for cancer [122,123,124,125,126]. The molecular anticancer effects and clinical trials for cancer using metformin have been recently reviewed by Buczyńska et al. [127] and Saraei et al. [128]. Metformin was also shown to reduce “chemobrain” [129]. Currently, the most pressing question is whether metformin might be effective and safe for patients without diabetes. A recent systematic review and meta-analysis found that, among patients without diabetes who were treated for breast cancer, metformin improved overall survival by 65%, compared to those who did not receive metformin [130]. However, another trial showed that 2000 mg of metformin increased the side effects and worsened the outcomes, compared to the conventional therapy for unresected, locally advanced non-small cell lung cancer [131]. Moreover, metformin showed no added benefit in a randomized phase II/III study that compared paclitaxel/carboplatin/metformin to paclitaxel/carboplatin/placebo, as initial therapy for stage III, IVA, IVB, or recurrent endometrial cancer [132]. Ongoing studies aim to identify the types of patients who might benefit from metformin. 

Metformin therapy has been found to be safe when used for other non-diabetic conditions including obesity [133], rheumatoid arthritis [134], and for women with metastatic breast cancer [135] although there was insufficient evidence of the benefits on survival. Limiting side effects of metformin are predominantly gastrointestinal. A careful review of potential drug reactions with planned chemotherapy or treatment is recommended. 

## 12. Statin Therapy

Statins were recognized in the 1990s as effective anticancer agents, but their use as adjuncts for patients with cancer remains controversial and consequently has not been approved as a cancer therapy. Research and in vivo studies have demonstrated mixed results. A provider considering adding a statin as adjunctive therapy should carefully review the most current literature regarding the chemotherapy agent that is being used and the possible adjunctive statin. This is an area that is receiving a great deal of attention and research.

An inverse relationship between statin use and the incidence of head and neck cancer was found in an analysis of a large population data base in Taiwan [136]. Bourguillon et al. concluded that statins may be beneficial by reducing side effects from chemotherapy/radiotherapy and improving survival based on the results of HNC trials [137]. A 2020 review by Matusewicz et al. [138] suggested that statins may be beneficial for treating metastatic cancers, due to their inhibition of the epithelial-mesenchymal transition. In this transition, cancer cells gain metastatic abilities and may become multipotent mesenchymal cancer-like stem cells. Statins have inhibited this activity in prostate, ovarian, breast, and esophageal cancers. In addition, because statins inhibit HMG CoA reductase, the rate- limiting step in cholesterol synthesis, they limit the production of mevalonate. Mevalonate produces coenzyme Q10, which is an important electron transporter in the electron transport chain. Statins will lower CoQ10 levels and cause abnormal mitochondrial respiration, loss of antioxidant protection and increase apoptotic cell death [139]. Mevalonate also gives rise to acetoacetyl-CoA, an activator of proinflammatory macrophages [140].

Seckl reported, however, that although pravastatin was well tolerated with standard chemotherapy treatment for small cell lung cancer, there was no benefit in outcome benefit [141]. Whether the statin is hydrophilic or lipophilic maybe important regarding their effectiveness against cancer [142]. A recent review by Pun elicits statins effects both as monotherapy against cancer and the potential synergistic effect with anticancer drugs to reduce drug resistance but also noted the excessive toxicity that has occurred in some situations [143]. The synergistic effect of lovastatin was confirmed by invitro analysis with tamoxifen, doxorubicin, methotrexate, and rapamycin but not with 5-fluoruracil, gemcitabine, epothilone, cisplatin, cyclophosphamide and etoposide [144]. A retrospective observation review found that the combination of statin and metformin therapy in a cohort of patients with prostate cancer was well tolerated and was associated with lower Gleason score and longer survival [145]. A careful review of potential drug reactions with planned chemotherapy or treatment is recommended. 

## 13. Conclusions

Providers also need to be aware of the potential benefit of non-traditional therapies to improve the patient’s outcomes with cancer. Although they may have been trained to use only medications or supplements that have been proven in rigorous scientific human trials, it may be time to become more open to including complementary or adjunctive therapies, particularly focusing on the risk/benefit ratio for each patient. Suggestions reviewed include the use of medications “off label” that could potentially help patients without causing harm. Patients need help in fighting cancer from each of their providers and may be more open to several of these recommendations and referrals if made by their practitioner. However, any adjunctive medication therapy considered should be carefully researched by the provider to help reduce harmful side effects while such recommendations as attention to a balance diet, increase exercise, optimized sleep, and regular mindfulness are usually of low risk.

## Figures and Tables

**Table 1 cancers-14-04780-t001:** Dietary recommendations from the World Cancer Research Fund and the American Institute for Cancer Research [62].

Increase:	Whole grains, fruits, vegetables, beans, nuts (goal is to exceed 30 gm fiber per day and 5 portions of fresh fruits/vegetables daily)
Reduce:	Fast foods and processed foods which may be high in simple carbohydrates and fat; sugary drinks, red meat (to maximum 3 portions/week) and avoid processed meats
Limit:	Consumption of alcohol
Overall goal	Reduce weight to normal BMI and avoid weight gain

**Table 2 cancers-14-04780-t002:** Major effects of green tea catechins on cancer cells (not all inclusive).

Induces:	Cellular apoptosis Cellular necrosis Cell cycle arrest
Impacts:	Cell morphology Protein synthesis
Inhibits:	Metastasis Angiogenesis Proliferation DNA methylation Immune checkpoint proteins Transcription and translation of genes that encode stemness markers Spheroid formation in stem cells Glutamine dehydrogenase and other enzyme pathways
Other effects	Anti-inflammatory Anti-oxidative and pro-oxidative effects

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
