# Peer review of "Review of Under-Recognized Adjunctive Therapies for Cancer"

_cancers, 2022, doi:10.3390/cancers14194780_

Round 1

Reviewer 1 Report

Summary: The purpose of this review is stated to be an overview of underappreciated or unappreciated adjunctive measures that may contribute to improvement of the quality of life or extend survival of patients suffering from cancer. It is relevant because it helps bringing the adjunctive modalities to the attention of the medical and scientific community.

General concepts/comments

Authors never state that this is not a comprehensive review, and never give a reason for the adjunctive therapies addressed in this review.

Review

Suggested review and changes:

Exercise:

-        Include the ASCO guidelines 1

Mindfulness/Meditation Practice

-        Consider removing the reference to the clinical trial NCT04800419, relevance?

-        The Sudarshan Kriya Yoga is effective in anxiety and depression but the citation provide was not of a study of this technique in cancer patients.

-        If they want to reference useful de-stress, anti-anxiety techniques in cancer, yoga, tai chi and qigong come to mind. These are not discussed in this review. I would suggest to add them.

Cancer and the Circadian Rhythm/Sleep Dysfunction

-        Review font in first paragraph

-        Review the whole section to make it relevant to the intention of the paper: adjunctive therapies during cancer treatment

Melatonin

-        Melatonin hormone regulates sleep-wake cycle, not sleep disorders, as stated by the authors.

-        This section is different from previous ones, having overview section and effects on cancer. Consider to have a uniform presentation for all modalities addressed

-        Under “Effects of melatonin on cancer” is confusing. The first paragraph is of a study using anti-cancer dose of melatonin, to see effect on outcome, while the second paragraph is melatonin implications for sleep with a study on breast cancer using a much lower dose of melatonin.

Diet / Smoking Cessation

-        There should be two distinct sections, one for Diet and one for smoking cessation.

-        Research in animals does not always translate with same effects to humans. For this review and for consistency throughout the entire paper, I recommend to avoid animal studies or cell studies.

Green Tea

-        How do we understand 300mg of GTE, are all extracts of GT the same?

-        Mechanism of GT effect on cancer cells: somehow this seems out of place. I would recommend to add the mechanism in relationship to the specific cancer, since is not given for any of the other adjunctive therapies.

-        Some mention of GT components for prevention, make sure it fits on this review

- Remove anecdotal patient with ovarian cancer. It is encouraging but does not belong on a review paper.

Curcumin/Turmeric: no corrections

Aspirin: no corrections

Metformin: no corrections

Statin Therapy: Revision of the last statement on this section is recommended.

The following pertains to the supplements and medicines mentioned on this article, and a statement or paragraph about the following should be added:

-The safety of the supplements and medicines taken with cancer treatment should be stated for each one of them. Clearly, some of them are not standard of care. In particular, related to newer anti-cancer treatment such as targeted and immunotherapies.

General: Structure each supplement medicine, and when possible all other modalities similarly: overview, mode of action, and human studies/therapeutic anticancer potential

Line 262 should say PROOF-OF-PRINCIPLE (instead of proof-of-principal)

One of Dr. Money’s patients permitted the reference of her ovarian 369 cancer therapy and CA 125 response to green tea in this document: this is better for a case report than on this review

Bibliography:

-        Some references are missing full last names of the authors

1-    Exercise, Diet, and Weight Management During Cancer Treatment: ASCO Guideline.

Ligibel JA, Bohlke K, May AM, Clinton SK, Demark-Wahnefried W, Gilchrist SC, Irwin ML, Late M, Mansfield S, Marshall TF, Meyerhardt JA, Thomson CA, Wood WA, Alfano CM.J Clin Oncol. 2022 Aug 1;40(22):2491-2507. doi: 10.1200/JCO.22.00687. Epub 2022 May 16.PMID: 35576506 Review.

Author Response

Summary: The purpose of this review is stated to be an overview of underappreciated or unappreciated adjunctive measures that may contribute to improvement of the quality of life or extend survival of patients suffering from cancer. It is relevant because it helps bringing the adjunctive modalities to the attention of the medical and scientific community.

General concepts/comments

Authors never state that this is not a comprehensive review, and never give a reason for the adjunctive therapies addressed in this review.

Please see the highlighted sentence in the brief summary.

Review

Suggested review and changes:

Exercise:

  •        Include the ASCO guidelines 1

This was added as suggested.  Thank you for recommending this addition. Please see the opening sentence in red in the Exercise section.

Mindfulness/Meditation Practice

-        Consider removing the reference to the clinical trial NCT04800419, relevance? This was removed. 

-        The Sudarshan Kriya Yoga is effective in anxiety and depression but the citation provide was not of a study of this technique in cancer patients.

  •        If they want to reference useful de-stress, anti-anxiety techniques in cancer, yoga, tai chi and qigong come to mind. These are not discussed in this review. I would suggest to add them.

This section was completely revised to incorporate Tai Chi, yoga, and Baduanjin as alternative options for exercise for patients.  And add another section on stress reduction techniques.  Please see the sections in red.

Cancer and the Circadian Rhythm/Sleep Dysfunction

  •        Review font in first paragraph This was corrected. 
  •        Review the whole section to make it relevant to the intention of the paper: adjunctive therapies during cancer treatment
  • Please see the first paragraph of this section regarding the reason it was included.

Melatonin

-        Melatonin hormone regulates sleep-wake cycle, not sleep disorders, as stated by the authors. This was corrected.

-        This section is different from previous ones, having overview section and effects on cancer. Consider to have a uniform presentation for all modalities addressed

-        Under “Effects of melatonin on cancer” is confusing. The first paragraph is of a study using anti-cancer dose of melatonin, to see effect on outcome, while the second paragraph is melatonin implications for sleep with a study on breast cancer using a much lower dose of melatonin.

The sections on sleep dysfunction and melatonin were both substantially revised to reflect how a low melatonin level was common after chemotherapy, and  the melatonin section focus was on the use as a chemotherapy agent.  Please see the red sections in each.

Diet / Smoking Cessation

-        There should be two distinct sections, one for Diet and one for smoking cessation. 2 sections were created.

-        Research in animals does not always translate with same effects to humans. For this review and for consistency throughout the entire paper, I recommend to avoid animal studies or cell studies. 

The section on animal studies and the keto diet was removed.  More studies in clinical trials were added.  See this section revision.

Green Tea

-        How do we understand 300mg of GTE, are all extracts of GT the same?

Not all GTE are the same and this was discussed.  The 300 mg of GTE that was referred to in the one prostate study was specially formulated.  Otherwise a comment has been added that not all GTE are the same. 

  •      Mechanism of GT effect on cancer cells: somehow this seems out of place. I would recommend to add the mechanism in relationship to the specific cancer, since is not given for any of the other adjunctive therapies. Unfortunately  there are very few studies reported regarding the effect of green tea since it is so difficult to standardize.  How can you be sure that a patient consumed 6-8 mugs of green tea, did they brew it correctly, etc.  However, the literature is quite strong on the effects that it has on specific cells and we felt it was important for the providers to have this knowledge and be able to discuss it with their patients. 

-        Some mention of GT components for prevention, make sure it fits on this review Again, the evidence of it's effect is strongest in the prevention data, and very limited in any clinical studies.  We felt it was additive to the recommendation for green tea to be considered.

  • Remove anecdotal patient with ovarian cancer. It is encouraging but does not belong on a review paper. This was removed.

Curcumin/Turmeric: no corrections

Aspirin: no corrections

Metformin: no corrections

Statin Therapy: Revision of the last statement on this section is recommended.

Please see the revision. 

The following pertains to the supplements and medicines mentioned on this article, and a statement or paragraph about the following should be added:

-The safety of the supplements and medicines taken with cancer treatment should be stated for each one of them. Clearly, some of them are not standard of care. In particular, related to newer anti-cancer treatment such as targeted and immunotherapies. A statement was added to this effect for metformin and statis.

General: Structure each supplement medicine, and when possible all other modalities similarly: overview, mode of action, and human studies/therapeutic anticancer potential. This was attempted and the last section of each was safety issues.

Line 262 should say PROOF-OF-PRINCIPLE (instead of proof-of-principal)

This was corrected.

One of Dr. Money’s patients permitted the reference of her ovarian 369 cancer therapy and CA 125 response to green tea in this document: this is better for a case report than on this review

This was removed.

Bibliography:

-        Some references are missing full last names of the authors This was corrected.

1-    Exercise, Diet, and Weight Management During Cancer Treatment: ASCO Guideline. Added.

Ligibel JA, Bohlke K, May AM, Clinton SK, Demark-Wahnefried W, Gilchrist SC, Irwin ML, Late M, Mansfield S, Marshall TF, Meyerhardt JA, Thomson CA, Wood WA, Alfano CM.J Clin Oncol. 2022 Aug 1;40(22):2491-2507. doi: 10.1200/JCO.22.00687. Epub 2022 May 16.PMID: 35576506 Review.

Reviewer 2 Report

This review emphasizes the significance of fitness and other underappreciated complementary therapies for cancer patients, including meditation or mindfulness, improved sleep hygiene, melatonin, turmeric, green tea, dietary adjustments, and repurposed pharmaceuticals. The evaluation also includes the "off-label" usage of drugs that may aid patients without causing harm. However, the authors also note that any adjunctive medication therapy considered should be thoroughly researched in order to reduce harmful side effects, whereas recommendations such as maintaining a balanced diet, increasing physical activity, optimising sleep, and practising mindfulness on a regular basis are typically low risk.

According to my assessment, it can be accepted for publication in the "Cancers".  

Author Response

Thank you for your comments.  I have expanded the manuscript to include additional exercise options and stress reduction techniques.  Please see the areas in red which are the revised sections. 

Round 2

Reviewer 1 Report

After reviewing the changes, I recommend this manuscript for publication